# The Triglyceride-Glucose Index Is Associated with Longitudinal Cognitive Decline in a Middle-Aged to Elderly Population: A Cohort Study

**DOI:** 10.3390/jcm11237153

**Published:** 2022-12-01

**Authors:** Siqi Li, Xuan Deng, Yumei Zhang

**Affiliations:** 1Department of Neurology, Beijing Tiantan Hospital, Capital Medical University, Beijing 100070, China; 2Clinical Research Institute, Shanghai General Hospital, Shanghai Jiao Tong University School of Medicine, Shanghai 200080, China; 3Department of Rehabilitation Medicine, Beijing Tiantan Hospital, Capital Medical University, Beijing 100070, China

**Keywords:** triglyceride-glucose index, cognitive decline, insulin resistance, risk factor, association

## Abstract

Background: To examine the effect of the triglyceride-glucose (TyG) index on longitudinal cognitive decline in a healthy middle-aged-to-elderly population. Methods: We conducted a population-based longitudinal study. A total of 1774 participants without cognitive impairment were enrolled in the 4-year follow-up. They were divided into four groups according to the quartile of the TyG index. Multivariable-adjusted Cox proportional hazard models were performed to examine the association between the TyG index and cognitive decline. Discrimination tests were used to evaluate the incremental predictive value of the TyG index beyond conventional risk factors. Results: During the follow-up, compared with those in the bottom quartile group, participants in the top TyG quartile group presented a 51% increase in the risk of cognitive decline (OR 1.51 (95% CI: 1.06–2.14)). As shown by discrimination tests, adding the TyG index into the conventional model resulted in a slight improvement in predicting the risk of cognitive decline (NRI 16.00% (*p* = 0.004)). Conclusion: This study demonstrated that increasing values of the TyG index were positively associated with the risk of cognitive decline. Monitoring the TyG index may help in the early identification of individuals at high risk of cognitive deterioration.

## 1. Introduction

With the aging of the global population, cognitive impairment is increasingly becoming a public health concern. Cognitive impairment, an aging-related disease, is clinically characterized by deterioration in memory, thinking, language, and executive function. Severe cognitive impairment may affect a person’s ability to perform daily activities and cause social and economic burdens [1,2]. The prevalence of dementia in China has increased by 5.6% in recent years, while the global prevalence has increased by 1.7% [3]. Many studies have shown that poor cognitive performance is associated with an increased risk of death, especially in middle-aged and elderly populations [4,5,6]. Research on risk factors for dementia has become a popular topic in recent years. Moreover, cognitive function changes over time. Therefore, it is necessary to identify and control the risk factors that may influence cognitive decline promptly.

Insulin resistance (IR) is defined as the reduced responsiveness of target tissue to insulin [7]. Previous studies have linked type 2 diabetes, caused by insulin resistance, to a higher risk of Alzheimer’s disease (AD) [8,9]. Other studies have found that increased serum insulin levels and insulin resistance are directly related to cognitive impairment [10,11,12]. The effect of this central IR on the occurrence and development of cognitive impairment may be related to changes in hippocampal synaptic plasticity [13]. Therefore, early detection and control of insulin resistance may be beneficial for the prevention of cognitive dysfunction. However, due to the complex testing process and high cost, the hyperinsulinemic-euglycemic clamp method is not commonly used in clinical settings to assess IR [14]. In addition, the degree of IR can be indexed by the homeostasis model assessment of insulin resistance (HOMA-IR) according to the homeostasis model assessment, which is calculated with the use of fasting glucose and insulin levels [15]. However, circulating insulin is not a routinely measured indicator, and this model has a high coefficient of variation due to the fluctuating pattern of insulin secretion, which makes HOMA-IR unsuitable for clinical diagnosis and large-scale studies [16].

The correlation between blood lipid levels and cognitive dysfunction has also received extensive amounts of attention. Previous studies have revealed that elevated blood lipid levels are linked to cognitive decline and the risk of AD [17,18,19,20,21]. Other studies have found that high lipid accumulation products are associated with cognitive performance and decline over four years, and this relationship may be influenced by sex and blood pressure [22,23]. Plasma triglycerides, simple lipids involved in energy storage and transportation, are thought to be closely associated with poor cognitive performance [24,25]. Increased plasma triglyceride levels may be involved in cognitive dysfunction through putative mechanisms such as blood–brain barrier disruption or an imbalance in amyloid metabolism [26,27,28,29]. Moreover, a previous study showed that elevated plasma triglyceride levels appear to be a risk factor for cognitive decline in elderly patients with diabetes [30]. Similarly, a cohort study also found that higher levels of triglycerides in midlife were associated with the decline in cognitive functions such as memory, attention, and executive function after 20 years of follow-up [20]. Another study explored whether triglyceride levels mediate cognitive decline in patients with major depressive disorder, and found that triglyceride levels are involved in the progression of depression and play an important role in memory decline [31].

Recently, the triglyceride-glucose (TyG) index calculated from fasting triglyceride and blood glucose has become easy to collect, cost-effective, and reliable, and has been suggested as a surrogate marker for evaluating insulin resistance [32,33]. Insulin resistance can be predicted better by the TyG index compared to the HOMA-IR index, according to researchers [34]. Previous studies have suggested that a high TyG index is associated with cardiovascular diseases, but population-based longitudinal studies on the association of the TyG index with cognitive impairment are lacking [35,36,37]. One study investigated the relationship between the TyG index and the burden of cerebral small vessel disease and cognitive impairment (CI) in elderly patients with diabetes and found a positive correlation between the increased TyG index and CI [38]. Another study revealed that the relationship between lung function and subsequent cognitive function in middle-aged and elderly people with a systemic low-grade inflammation state could be mediated by the TyG index [39]. Moreover, another study found that the TyG index was associated with a higher risk of dementia [40]. However, the relationship between the TyG index and cognitive decline remains poorly understood. Therefore, in this study, we aimed to characterize the association between the TyG index and longitudinal cognitive decline in healthy middle-aged and elderly populations.

## 2. Materials and Methods

### 2.1. Study Population

The Jidong Cognitive Impairment Cohort Study (CICS) is a community-based, long-term observational cohort study designed to investigate the potential prognostic factors and the risk of cognitive impairment [41]. The inclusion criteria for participants in this study include the following: (i) age of 40 years or older; (ii) no related diseases that may affect the cognitive function assessments, for instance, severe aphasia, hearing loss, visual impairment, psychosis or schizophrenia (documented in the questionnaire); and (iii) provision of signed informed consent. A total of 3617 participants aged 40 years or older were recruited into the CICS in 2015. Among these participants, the following were further excluded: 75 individuals without fasting blood glucose and triglyceride data in 2015, 29 individuals without an education record, 108 individuals with cognitive impairment in 2015, 932 individuals without cognitive examinations in 2015 and 2019, and 699 participants who were lost during follow-up. Eventually, a total of 1774 participants were ultimately enrolled in this study and were followed for 4 years until 2019.

The present study was performed in accordance with the guidelines described by the Helsinki Declaration and was approved by the Ethics Committees of Kailuan General Hospital of Tangshan City and the Medical Ethics Committee, the Staff Hospital of Jidong Oilfield Branch, China National Petroleum Corporation (No. 2013 YILUNZI 1). All the participants signed a written informed consent form prior to their inclusion in the study.

### 2.2. Data Collection

The information from all participants, including demographic characteristics and clinical examination information (age, sex, body mass index (BMI), smoking, alcohol consumption, educational level, and physical activity) and medical history (hypertension, dyslipidemia, diabetes mellitus), was collected through a series of comprehensive questionnaires. Regular physical activity was defined as four or more times per week of exercise. BMI was calculated as weight in kilograms divided by the square of height in meters. Hypertension was defined as a self-reported history, any current use of antihypertensive medications, or a diagnosis of hypertension during healthcare examination (systolic blood pressure ≥ 140 mmHg, or diastolic blood pressure ≥ 90 mmHg). Diabetes mellitus was defined as a self-reported history, any current use of hypoglycemic medications, or fasting blood glucose levels ≥ 7.0 mmol/L. Hyperlipidemia was defined as a self-reported history, any current use of lipid-lowering therapy, or a diagnosis of hyperlipidemia during healthcare examination (serum triglyceride ≥ 1.7 mmol/L, total cholesterol ≥ 5.72 mmol/L, high-density lipoprotein ≤ 0.9 mmol/L). In addition, fasting blood glucose (FBG), total cholesterol (TC), triglycerides (TGs), high-density lipoprotein (HDL), and low-density lipoprotein (LDL) levels from blood samples were collected carefully from the antecubital vein in the morning under fasting conditions. Fasting blood glucose was measured using the hexokinase/glucose-6-phosphate dehydrogenase method, and triglyceride concentrations were determined by enzymatic methods (Mind Bioengineering Co., Ltd., Shanghai, China). All the blood samples were stored in serum-separated tubes and EDTA anticoagulant collection tubes at −80 °C to address potential sources of bias.

### 2.3. Assessment of the TyG Index

The TyG index was calculated as ln (fasting triglyceride (mg/dL) × fasting glucose (mg/dL)/2), as previously described [32,33].

### 2.4. Outcome Evaluation

All participants in this study completed the Chinese version of the Mini-Mental State Examination (MMSE) to evaluate cognitive function. The MMSE consists of 30 items that assess five cognitive domains: orientation (10 points), regulation (3 points), attention and calculation (5 points), recall (3 points) and language (10 points). Cognitive impairment was defined as the education-based cutoffs of MMSE scores: ≤17 for illiterate individuals, ≤20 for primary school graduates, and ≤24 for junior high school graduates or above. This cutoff point has been shown to have the best sensitivity and specificity in the Chinese population [42]. Generally, relatively high scores represent increased cognitive functioning.

### 2.5. Statistical Analysis

The data were tested for normality by using the Kolmogorov–Smirnov test. Continuous variables are presented as the means ± standard deviations for normally distributed data and were compared using ANOVA. Continuous variables that did not exhibit a normal distribution are presented as medians with interquartile ranges and were compared by the Kruskal–Wallis U test. Categorical variables were expressed as frequencies (proportions) and were compared using the χ^2^ test or Fisher’s exact test.

Participants in this study were divided into four categories according to the quartile of the TyG index, which was assessed at baseline. The association of the TyG index with cognitive decline was estimated by the use of the multivariate Cox proportional hazard regression model. Data reported as hazard ratios (HRs) and 95% confidence intervals (CIs) were calculated after we adjusted for potential confounding factors. Model 1 was adjusted for age, sex and educational levels. Model 2 was additionally adjusted for smoking status, alcohol consumption, physical activity, body mass index, history of hypertension, and total cholesterol. To avoid collinearity, hypertension rather than systolic and diastolic blood pressure, BMI but not waist circumference, and total cholesterol but not HDL and LDL were used in the model fitting. In addition, restricted cubic spline analysis was used to address the dose–response relationship between the TyG index and cognitive decline. Furthermore, discrimination tests (C statistic, net reclassification improvement (NRI) and integrated discrimination improvement (IDI)) were used to evaluate the incremental predictive value of the TyG index beyond conventional risk factors. Additionally, subgroup analyses were conducted after stratification by age (<60 y or ≥60 y), sex, BMI (<25 or ≥25 kg/m^2^), hypertension, dyslipidemia and diabetes mellitus to test interactions and assess whether the effect of the TyG index on cognitive decline differed between different subgroups, which was adjusted by model 2. 

In this study, *p* values < 0.05 were considered to be statistically significant and were two-sided.

All the statistical analyses were conducted using SAS version 9.4 (SAS Institute Inc., Cary, NC, USA).

## 3. Results

### 3.1. Demographic and Clinical Characteristics

Among 3617 participants aged 40 or older in the CICS, 1774 participants who met the inclusion criteria were finally enrolled (Figure 1), and their baseline characteristics are shown in Table 1 and Appendix A. Their mean age was 53.48 ± 8.47 years, 52.03% were women, and 90.98% had a junior high school educational level or higher. The participants were categorized into four groups according to the quartile of the TyG index at baseline (<7.87, 7.87–8.25, 8.25–8.68, ≥8.68). Compared with participants in the Q1 group, we found that patients with a higher TyG index were predominantly older, men, obese, more current smokers and drinkers, had a higher prevalence of hypertension, diabetes mellitus, and dyslipidemia, had a high FBG, TC, TG, LDL level and had a low HDL level.

### 3.2. Effect of the Tyg Index on Cognitive Decline

Table 2 shows the cognitive performance of participants with different TyG quartiles in 2015 and 2019. There was no significant difference in cognitive performance between these four groups in 2015 (28.51 ± 1.49 in Q1 vs. 28.46 ± 1.64 in Q4, *p* = 0.742). In 2019, participants in the highest TyG level group had worse cognitive performance (27.80 ± 2.79 in Q1 vs. 27.24 ± 3.12 in Q4, *p* = 0.002) and faster cognitive decline from 2015 to 2019 (0.70 ± 2.83 in Q1 vs. 1.21 ± 3.30 in Q4, *p* = 0.009) than those in the lowest TyG level group.

During the 4-year follow-up period from 2015 to 2019, among these 1774 participants, 820 (46.22%) participants presented cognitive decline. Associations of TyG levels with cognitive decline are presented in Table 3. After adjusting for the potential confounding factors mentioned above, the odds ratio for cognitive decline increased across TyG quartiles: 1.17 (95% CI: 0.85–1.62), 1.31 (95% CI: 0.93–1.83), and 1.51 (95% CI: 1.06–2.14) for the 2nd, 3rd, 4th quartiles, respectively, using the 1st quartile as the reference (*p* for trend = 0.020). Compared with those in the lowest TyG group, participants in the highest TyG group recorded a 51% increase in the risk of cognitive decline. Notably, the restricted cubic spline analysis also demonstrated that a higher TyG index represented a higher risk of cognitive decline (Figure 2).

After applying discrimination tests, we found that there was a slight improvement in predicting the risk of cognitive decline when adding the TyG index to the conventional model (including age, sex, educational levels, smoking status, alcohol consumption, physical activity, BMI, history of hypertension, total cholesterol) (NRI 16.00% (*p* = 0.004), IDI 0.40% (*p* = 0.030)) (Table 4).

According to previous studies, several demographic and physiological factors might cause different effects of the TyG index on cognitive decline [40,41,42,43]. Thus, we further analyzed the interaction effects in this study. Regardless of the stratification of age, sex, BMI, hypertension, dyslipidemia and diabetes mellitus, participants in the highest quartile of the TyG index had a higher risk of cognitive decline than those in the lowest quartile. This association was significant in the female group (*p* value for interaction = 0.028). Moreover, the analysis showed no significant interaction effects between the TyG index and other stratified variables, including age, BMI, hypertension, dyslipidemia and diabetes mellitus (*p* value for interaction effects > 0.05) (Appendix A).

## 4. Discussion

This longitudinal cohort study revealed a positive association between the TyG index and the risk of cognitive decline. Compared with those in the bottom of the fourth quartile, participants in the top of the fourth quartile showed a 51% increased risk of cognitive decline. The risk of this outcome increased with the increasing TyG quartile regardless of age, sex, educational levels, smoking status, alcohol consumption, physical activity, BMI, history of hypertension, and TC. Furthermore, adding the TyG index into the conventional model promoted the risk stratification ability. In addition, we found that the effect of the TyG index on the risk of cognitive decline remained consistent across the subgroup analyses, but this positive correlation was more significant among women.

In the past, the relationship between insulin resistance and dementia has been investigated; however, the HOMA-IR index has been mainly used in studies to determine the degree of IR; however, the HOMA-IR index values are cumbersome to collect and have low possibility of clinical application. Previous studies have reported that a higher risk of dementia was associated with increased insulin levels and HOMA-IR index [10,12,44,45,46,47]. In populations at high risk of cardiovascular disease, baseline HOMA-IR levels are associated with short-term cognitive deterioration [48]. Conversely, some researchers have found no association between dementia and insulin resistance [49]. Recently, a study found that elevated levels of the TyG index were associated with the risk of dementia and suggested that this index, as a marker of IR, might be a potential predictor of dementia development [40]; however, the evidence is still insufficient. The strength of this cohort study is that it further explored the effect of the TyG index, a surrogate marker of insulin resistance, on longitudinal cognitive function decline in a healthy middle-aged to elderly Chinese population.

The results of this current study could be explained by the following possible mechanisms. Previous studies have shown that insulin metabolism and hemodynamic actions of insulin share common intracellular transduction pathways [50]. Insulin resistance might play an important role in arterial stiffness and the formation of atherosclerotic plaque by causing chronic inflammation, oxidative stress, impaired nitric oxide activity, and endothelial dysfunction, promoting foam cell formation, altering estrogen receptor expression and further leading to hemodynamic changes [51,52]. As a receptor expressed in blood vessels and adipose tissue, the mineralocorticoid receptors (MR) can regulate adipose tissue function and vascular tone. Studies have shown that the presence of metabolic abnormalities such as obesity, insulin resistance, and increased aldosterone levels, favors MR activation, which further promotes vascular endothelial dysfunction and accelerates arterial stiffness and atherosclerosis [53,54]. Arterial stiffness has gradually become a proxy of arterial aging, reflecting the state of vascular health. A negative correlation has been found between arterial stiffness and cognition over time, independently of other risk factors. The predictive value of arterial stiffness as a risk marker for cognitive deterioration and dementia is independent of age and traditional cardiovascular risk factors [55,56,57]. Moreover, long-term insulin resistance prevents insulin from crossing the blood–brain barrier, leading to decreased insulin levels in the brain, which in turn impairs neuronal function and synaptic building [58]. Moreover, peripheral insulin resistance can weaken the insulin signal in the brain, decrease the uptake of brain plasma insulin and reduce the clearance of brain Aβ [59]. Recent studies have shown that insulin resistance might enhance amyloid-β production or its neurotoxicity and might also be involved in tau phosphorylation and neurofibrillary tangle formation [60]. One study found that in cognitively normal individuals, increasing levels of HOMA-IR were associated with increases in CSF T-tau and P-tau [61]. Another study reported that higher insulin resistance might predict the degree of amyloid substance deposition in the temporal and frontal lobes of patients with Alzheimer’s disease [62]. Severe IR promotes amyloid Aβ production, increases AD-type amyloid plaque burden, and may cause hippocampal atrophy and impair memory task performance. Areas of the brain with high concentrations of insulin receptors, such as the medial temporal lobe and frontal lobe, were reported to be particularly sensitive to insulin signals and especially more vulnerable in AD patients [63]. Studies have found that higher IR levels were associated with a lower cerebral glucose metabolic rate in these areas in adults with or without AD [62,64]. Decreased glucose metabolism might affect cortical activity, resulting in impaired cognitive function. In other words, insulin resistance accelerates cognitive deterioration by substantially affecting hippocampal plasticity, changes in amyloid precursor protein metabolism, increased tau protein concentration, altered brain inflammation, and the involvement of the ApoEε4 allele [61,63,65,66].

Moreover, the TyG index is calculated from fasting triglycerides and blood glucose. Triglycerides may affect cognitive function by penetrating the blood–brain barrier and inducing central insulin receptor resistance [28,67]. In addition, elevated triglyceride levels and blood glucose levels may reflect increased neuroinflammation, which may also affect cognitive function [68]. Alternatively, during the development of insulin resistance, insulin cannot inhibit the production of glucose in the liver but paradoxically accelerates the synthesis of lipids, leading to an increase in triglycerides and blood glucose [69]. Moreover, as insulin resistance leads to reduced uptake of glucose by nerve cells, the energy supply of neurons decreases, leading to increased oxidative stress, neuroinflammation, and lipid dysregulation, all of which in turn accelerate neurodegeneration.

In this study, we also observed sex differences in the association of the TyG index and cognitive decline, suggesting that the TyG index might be a risk factor for cognitive decline in women. Similar results have been found in previous studies [70,71]. One study found that insulin resistance was more prevalent in women than in men [61]. Metabolic syndrome was associated with the risk of cognitive decline in women compared with men [72]. These findings might be due to changes in sex hormone levels in middle-aged and elderly women. Moreover, as women age, the decrease in estrogen production diminishes its neuroprotective effects, which may lead to pathological changes associated with cognitive impairment.

This study also had several limitations. First, due to the shortage of insulin concentration data, we could not compare the correlation between cognitive function decline and the TyG index and HOMA-IR index. Second, the effect size of the TyG index on cognitive decline was relatively small, possibly due to the influence of sample size and follow-up time, and this might have underestimated the association between the TyG index and cognitive function decline. Thus, for the interesting results of this study to be of greater breadth and relevance, longer follow-up should be considered. Third, owing to the limitations of this observational study, we could not establish a causal link between the TyG index and the risk of cognitive decline. Furthermore, this study collected only baseline data of the TyG index; nevertheless, as a marker that reflects metabolism, it may undergo dynamic changes. Thus, the effect of dynamic changes in the TyG index on cognitive deterioration needs to be further verified.

## 5. Conclusions

In conclusion, this study found that an increased TyG index was positively associated with the risk of cognitive decline; moreover, adding the TyG index to the conventional model might provide superior risk stratification. As a readily measurable, cost-effective, and reliable marker for evaluating insulin resistance, the TyG index has potential clinical predictive value in predicting longitudinal cognitive decline, but its use still needs to be further confirmed. This finding highlights the importance of more effectively monitoring the TyG index in healthy middle-aged and elderly populations to identify individuals who are at high risk for cognitive impairment. More medical and lifestyle interventions should be performed as early as possible to prevent the development of cognitive deterioration in people with a high TyG index.

## Figures and Tables

**Figure 1 jcm-11-07153-f001:**
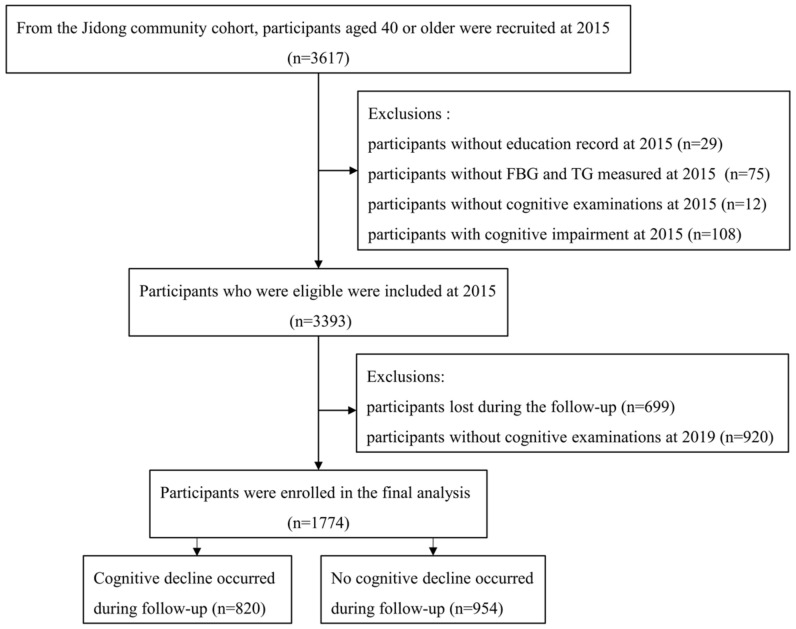
Flowchart of this study. Abbreviations: FBG, fasting blood glucose; TG, triglycerides.

**Figure 2 jcm-11-07153-f002:**
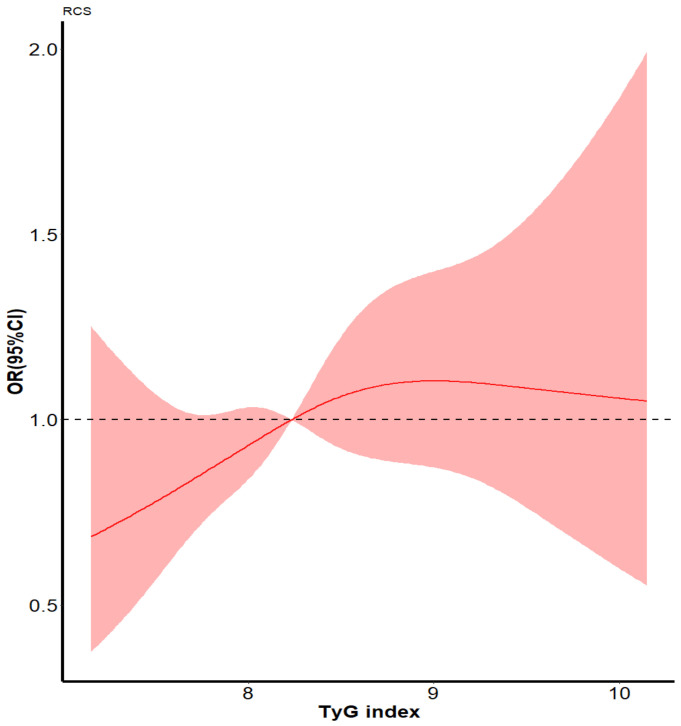
Spline models of the association between the TyG index and the risk of cognitive decline. The odds ratios from the multivariate Cox proportional hazard regression model were adjusted for the variables in model 2 in Table 3. The red line indicates the adjusted odds ratio, and the red area indicates the 95% confidence interval. Abbreviations: TyG, triglyceride-glucose.

**Table 1 jcm-11-07153-t001:** Baseline characteristics in 2015 of these 1774 participants according to quartiles of the TyG index.

Characteristic	Total	TyG Index				*p* Value
		Q1 (<7.87)	Q2 (7.87–8.25)	Q3 (8.25–8.68)	Q4 (≥8.68)	
N, (%)	1774	437	450	444	443	
Age, year, (mean ± SD)	53.48 ± 8.47	51.91 ± 8.58	53.59 ± 8.57	54.48 ± 8.47	53.92 ± 8.09	<0.001
Sex, n (%)						<0.001
Male	851(47.97)	148(33.87)	198(44.00)	250(56.31)	255(57.56)	
Female	923(52.03)	289(66.13)	252(56.00)	194(43.69)	188(42.44)	
Educational level, n (%)						0.601
Illiterate	65(3.66)	12(2.75)	17(3.78)	22(4.95)	14(3.16)	
Primary	95(5.36)	21(4.81)	28(6.22)	22(4.95)	24(5.42)	
Junior or above	1614(90.98)	404(92.45)	405(90.00)	400(90.09)	405(91.42)	
Body mass index, kg/m^2^, (mean ± SD)	25.02 ± 5.39	23.55 ± 9.22	24.52 ± 3.03	25.57 ± 2.99	26.43 ± 3.18	<0.001
Current smoking, n (%)	394(22.21)	66(15.10)	81(18.00)	114(25.68)	133(30.02)	<0.001
Current drinking, n (%)	553(31.78)	101(23.54)	121(27.38)	151(34.55)	180(41.67)	<0.001
Regular physical activity, n (%)	883(60.90)	210(57.69)	222(61.16)	239(66.02)	212(58.73)	0.489
Medical history, n (%)						
Hypertension	632(35.63)	87(19.91)	135(30.00)	179(40.32)	231(52.14)	<0.001
Diabetes mellitus	226(12.74)	10(2.29)	25(5.56)	56(12.61)	135(30.47)	<0.001
Dyslipidemia	1012(57.05)	60(13.73)	132(29.33)	382(86.04)	438(98.87)	<0.001
Laboratory test, (mean ± SD)						
TyG index	8.31 ± 0.62	7.60 ± 0.21	8.06 ± 0.11	8.45 ± 0.12	9.14 ± 0.44	<0.001
FBG, mg/dL	6.21 ± 1.42	5.67 ± 0.49	5.90 ± 0.72	6.17 ± 0.96	7.08 ± 2.29	<0.001
LDL, mg/dL	3.42 ± 0.82	3.01 ± 0.63	3.38 ± 0.79	3.61 ± 0.82	3.66 ± 0.86	<0.001
HDL, mg/dL	1.27 ± 0.27	1.43 ± 0.27	1.32 ± 0.25	1.21 ± 0.24	1.12 ± 0.23	<0.001
TC, mg/dL	5.15 ± 0.98	4.71 ± 0.78	5.09 ± 0.91	5.29 ± 0.95	5.51 ± 1.06	<0.001
TG, mg/dL	2.00 ± 1.53	0.91 ± 0.17	1.36 ± 0.19	1.94 ± 0.31	3.79 ± 2.09	<0.001

Abbreviations: Q, quartile; TyG, triglyceride-glucose; FBG, fasting blood glucose; TC, total cholesterol; TG, triglycerides; HDL, high-density lipoprotein; LDL, low-density lipoprotein levels.

**Table 2 jcm-11-07153-t002:** Cognitive function in 2015 and 2019 in participants according to the TyG quartiles.

	Overall	Q1	Q2	Q3	Q4	*p* Value
MMSE in 2015 (mean ± SD)	28.45 ± 1.58	28.51 ± 1.49	28.43 ± 1.62	28.39 ± 1.56	28.46 ± 1.64	0.742
MMSE in 2019 (mean ± SD)	27.38 ± 3.06	27.80 ± 2.79	27.48 ± 2.99	27.03 ± 3.30	27.24 ± 3.12	0.002
MMSE decline from 2015 to 2019 (mean ± SD)	1.06 ± 3.08	0.70 ± 2.83	0.95 ± 2.95	1.35 ± 3.18	1.21 ± 3.30	0.009
Cognitive decline incidence N (%)	820 (46.22)	180 (41.19)	198 (44.00)	218 (49.10)	224 (50.56)	0.017

Abbreviations: TyG, triglyceride-glucose; MMSE, Mini-Mental State Examination.

**Table 3 jcm-11-07153-t003:** ORs and 95% CIs for the association between the TyG index and the clinical outcome.

Outcome	TyG Levels				*p* for Trend
	Q1	Q2	Q3	Q4	
Cognitive decline from 2015 to 2019					
Events, N(%)	180 (41.19)	198 (44.00)	218 (49.10)	224 (50.56)	
Unadjusted	1	1.12 (0.86–1.46)	1.38 (1.06–1.80)	1.46 (1.12–1.91)	0.002
Model 1	1	1.07 (0.82–1.40)	1.30 (0.99–1.70)	1.40 (1.07–1.84)	0.006
Model 2	1	1.17 (0.85–1.62)	1.31 (0.93–1.83)	1.51 (1.06–2.14)	0.020

Abbreviation: TyG, triglyceride-glucose. Odds ratios (ORs) with 95% confidence intervals (CIs) were expressed by using multivariate Cox regression models. The OR of the 1st quartile was set as the reference. Model 1: adjusted for age, sex, and educational levels. Model 2: adjusted for the factors in model 1 plus smoking status, alcohol consumption, physical activity, body mass index, history of hypertension, and total cholesterol.

**Table 4 jcm-11-07153-t004:** Reclassification and disclination statistics for the clinical outcome when adding to the TyG index.

Outcome	Model	C-Statistic		NRI		IDI	
		Estimate (95% CI)	*p* Value	Estimate (95% CI)	*p* Value	Estimate (95% CI)	*p* Value
Cognitive decline from 2015 to 2019	Conventional model	0.64 (0.61–0.67)	0.414	Ref.	0.004	Ref.	0.030
	Conventional model +TyG	0.65 (0.62–0.68)		0.16 (0.05–0.27)		0.004 (0.003–0.01)	

Abbreviation: TyG, triglyceride-glucose; NRI, net reclassification improvement; IDI, integrated discrimination improvement; CI, confidence interval. Conventional model: added to factor-adjusted models, including age, sex, educational levels, smoking status, alcohol consumption, physical activity, body mass index, history of hypertension, and total cholesterol.

## Data Availability

The data used and/or analyzed in the current study are available on reasonable request from the corresponding author.

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
