# Peer review of "The Triglyceride-Glucose Index Is Associated with Longitudinal Cognitive Decline in a Middle-Aged to Elderly Population: A Cohort Study"

_jcm, 2022, doi:10.3390/jcm11237153_

Round 1

Reviewer 1 Report

I believe that this is an interesting research interrelating risk factors like insulin resistance with cognitive decline (CD) and analyzing a potential tool for the evaluation of the CD prognosis.  

However I have 3 comments.

1.     The authors should amplify the introduction with more related references and should also add a paragraph analyzing the impact of lipid levels and especially triglycerides in the CD since they are the main factor in TyG index.

2.     The authors should make a thorough check in the text for potential misspelling and typographical errors.

1.   Last but not least, I am wondering what was the authors contribution in Jidong Cognitive Impairment Cohort Study (CICS) with reference No 23, since their names are not included in this study. How they obtained the results and what was their involvement in the experiments concerning the results. If they are not related to this study they should have permission for using this study and clearly state it in the manuscript.

Author Response

We appreciate you for your carefulness and conscientiousness. We have studied your comments carefully, and have tried our best to improve and made the following revisions to this manuscript.

Point 1: The authors should amplify the introduction with more related references and should also add a paragraph analyzing the impact of lipid levels and especially triglycerides in the CD since they are the main factor in TyG index.

Response 1:Thank you for your constructive comments. We agree with your suggestion and we consulted more relevant references in the introduction part, which involved studies about the relationship between insulin resistance, the TyG index and cognitive impairment (lines 50-98, pages 2-3 of the revised manuscript). What’s more, we further analyzed the impact of blood lipid levels, especially plasma triglycerides in cognitive decline, with more related references (page 2 of the revised manuscript, lines 65-81).

Point 2: The authors should make a thorough check in the text for potential misspelling and typographical errors.

Response 2:Thank you for your suggestion. We are sorry for these typographic and grammar errors in the original article. We have corrected them. What’s more, the revised manuscript has been proofread and edited for language clarity and grammar by professional editing services (see the editing certificate in the attachment).

Point 3: Last but not least, I am wondering what was the authors contribution in Jidong Cognitive Impairment Cohort Study (CICS) with reference No 23, since their names are not included in this study. How they obtained the results and what was their involvement in the experiments concerning the results. If they are not related to this study they should have permission for using this study and clearly state it in the manuscript.

Response 3:We appreciate you very much for your question. The authors of the reference No.42 (The original 23rd reference) were also involved in the construction of the Jidong cognitive impairment cohort, such as the recruitment of subjects. This study is permitted by them. Moreover, no conflict of interest exist in the submission of this manuscript, and the manuscript is approved for publication. We especially thank these people for their contribution in the acknowledgments section (lines 378-381, page 11 of the revised manuscript).

We would like to thank you again for taking the time to review our manuscript. We have studied comments carefully and have made corrections which we hope meet with approval!

Reviewer 2 Report

1. The division of the extract into background, methods, results, and conclusions is redundant. See the guidelines for authors:

Abstract: The abstract should be a single paragraph and should follow the style of structured abstracts but without headings: 1) Background: Place the question addressed in a broad context and highlight the purpose of the study; 2) Methods: Describe briefly the main methods or treatments applied. Include any relevant preregistration numbers and species and strains of any animals used. 3) Results: Summarize the article's main findings; and 4) Conclusion: Indicate the main conclusions or interpretations. The abstract should be an objective representation of the article: it must not contain results that are not presented and substantiated in the main text and should not exaggerate the main conclusions.

2. Please explain why you describe in M&M that your study was performed and was approved by Jidong Oilfield Branch, China National Petroleum Corporation (No. 2013 YILUNZI 1). What does Oilfield & Petroleum Corporation have in common with this type of research?

The χ2 test belongs to the group of nonparametric tests. Its algorithm is based on comparing the frequency of events resulting from (empirical) experiences with those expected. The null hypothesis H0 assumes that the distribution of the variable we are investigating is consistent with the given theoretical distribution. What model was adopted as the reference value for testing the significance of the obtained results?

chapter 2.1 Study population. In this chapter, you explain how the study population was constructed. For me, that information is unclear. You describe that "3,617 participants aged 40 years or older who were recruited into the CICS in 2015 and were followed for 4 years until 2019." This information is explained once again in Figure 1 in Results, and in the scheme, there is only information about 2015, not 2019. Is this was the same group of patients examined twice (2015, 2019) or two independent examinations? there is no any explanation in the description of figure 1, table 1 & table 3, and table 4 (which patients are here?); on the other hand, in figure 2 there is a description of 2015 and 2019.

The course of the experiment and its stages are unclear to me. Perhaps the diagram of its stages over time should be better presented.

Author Response

We greatly appreciate the time and effort you have put into this manuscript. We have considered your suggestion carefully and have tried our best to improve this manuscript.

Point 1: The division of the extract into background, methods, results, and conclusions is redundant. See the guidelines for authors.

Response 1:We appreciate you very much for your suggestion, and we indeed should have improved it in the revised manuscript according to your advice and the guidelines for authors. In the part of background, we highlighted the purpose of the study. In the part of methods, we describes the main methods applied briefly. In the part of results and conclusion, we edited out the redundant part and objectively summarized the study's main findings and interpretations (lines 12-34, page 1).  

Point 2: Please explain why you describe in M&M that your study was performed and was approved by Jidong Oilfield Branch, China National Petroleum Corporation (No. 2013 YILUNZI 1). What does Oilfield & Petroleum Corporation have in common with this type of research?

Response:Thank you for your comment extremely. In this study, we recruited the participants in the staff hospital of Oilfield & Petroleum Corporation.And our study was approved by the Medical Ethics Committee of the staff hospital of Jidong Oilfield Branch, China National Petroleum Corporation (No. 2013 YILUNZI 1).

The χ2 test belongs to the group of nonparametric tests. Its algorithm is based on comparing the frequency of events resulting from (empirical) experiences with those expected. The null hypothesis H0 assumes that the distribution of the variable we are investigating is consistent with the given theoretical distribution. What model was adopted as the reference value for testing the significance of the obtained results?

Response:We appreciate your question. In this study, the alpha value was set as 0.05. The smaller the value of alpha, the less likely it is that we reject a true null hypothesis. We also emphasized it in the revised manuscript. “P values<0.05 were considered to be the reference value for testing the significance of the obtained results” (see lines 185-186, page 4).

chapter 2.1 Study population. In this chapter, you explain how the study population was constructed. For me, that information is unclear. You describe that "3,617 participants aged 40 years or older who were recruited into the CICS in 2015 and were followed for 4 years until 2019." This information is explained once again in Figure 1 in Results, and in the scheme, there is only information about 2015, not 2019. Is this was the same group of patients examined twice (2015, 2019) or two independent examinations? there is no any explanation in the description of figure 1, table 1 & table 3, and table 4 (which patients are here?); on the other hand, in figure 2 there is a description of 2015 and 2019. The course of the experiment and its stages are unclear to me. Perhaps the diagram of its stages over time should be better presented.

Response:Thanks for your advisement extremely. In this cohort study, it is the same group of these participants examined twice (2015, 2019). We are sorry that we did not clarify this issue in the previous manuscript. We further improved the main document and redrawn the flowchart according to your suggestion (lines 101-115 of page 3, lines 205-206 of page 6, lines 221-222 of page 7, revised Figure 1). We firstly identified 3,617 participants aged 40 years or older who were recruited into the CICS at 2015. Among these participants, we further excluded 29 subjects without an education record at 2015, 75 subjects without fasting blood glucose and triglyceride data at 2015, 108 subjects with cognitive impairment at 2015, 932 subjects without cognitive examinations at 2015 and 2019, 699 subjects who were lost during the follow-up. Eventually, a total of 1,774 participants were finally enrolled in our study, and were followed for 4 years until 2019.

We appreciate for your warm work earnestly, and hope that the correction in the revised manuscript will meet with approval. Once again, thank you very much for your suggestion!

Round 2

Reviewer 2 Report

Corrections have been made carefully.

Please check - figure 1 appears to occur twice.

Author Response

We greatly appreciate the time and effort you have put into this manuscript. We have considered your suggestion carefully and improve this manuscript.

Point 1: Please check - figure 1 appears to occur twice.

Response 1:Thank you for your comment extremely. We are sorry that Figure 1 appeared twice in the previous manuscript. We further improved it according to your suggestion (lines 187-190 of page 5, revised Figure 1).

We appreciate for your warm work earnestly, and hope that the correction in the revised manuscript will meet with approval!
